# Performance of a Shotgun Prediction Model for Colorectal Cancer When Using 16S rRNA Sequencing Data

**DOI:** 10.3390/ijms25021181

**Published:** 2024-01-18

**Authors:** Elies Ramon, Mireia Obón-Santacana, Olfat Khannous-Lleiffe, Ester Saus, Toni Gabaldón, Elisabet Guinó, David Bars-Cortina, Gemma Ibáñez-Sanz, Lorena Rodríguez-Alonso, Alfredo Mata, Ana García-Rodríguez, Victor Moreno

**Affiliations:** 1Colorectal Cancer Group, ONCOBELL Program, Institut de Recerca Biomedica de Bellvitge (IDIBELL), L’Hospitalet de Llobregat, 08908 Barcelona, Spain; 2Unit of Biomarkers and Suceptibility (UBS), Oncology Data Analytics Program (ODAP), Catalan Institute of Oncology (ICO), L’Hospitalet del Llobregat, 08908 Barcelona, Spain; 3Consortium for Biomedical Research in Epidemiology and Public Health (CIBERESP), 28029 Madrid, Spain; 4Barcelona Supercomputing Centre (BSC-CNS), 08034 Barcelona, Spain; 5Institute for Research in Biomedicine (IRB Barcelona), The Barcelona Institute of Science and Technology, 08028 Barcelona, Spain; 6Catalan Institution for Research and Advanced Studies (ICREA), 08010 Barcelona, Spain; 7Centro de Investigación Biomédica En Red de Enfermedades Infecciosas (CIBERINFEC), 08028 Barcelona, Spain; 8Gastroenterology Department, Bellvitge University Hospital, L’Hospitalet de Llobregat, 08907 Barcelona, Spain; 9Digestive System Service, Moisés Broggi Hospital, 08970 Sant Joan Despí, Spain; 10Endoscopy Unit, Digestive System Service, Viladecans Hospital-IDIBELL, 08840 Viladecans, Spain; 11Department of Clinical Sciences, Faculty of Medicine and Health Sciences, Universitat de Barcelona Institute of Complex Systems (UBICS), University of Barcelona (UB), L’Hospitalet de Llobregat, 08908 Barcelona, Spain

**Keywords:** colon cancer, gut microbiota, shotgun, 16S, metagenomics, predictive model, microbial signature

## Abstract

Colorectal cancer (CRC), the third most common cancer globally, has shown links to disturbed gut microbiota. While significant efforts have been made to establish a microbial signature indicative of CRC using shotgun metagenomic sequencing, the challenge lies in validating this signature with 16S ribosomal RNA (16S) gene sequencing. The primary obstacle is reconciling the differing outputs of these two methodologies, which often lead to divergent statistical models and conclusions. In this study, we introduce an algorithm designed to bridge this gap by mapping shotgun-derived taxa to their 16S counterparts. This mapping enables us to assess the predictive performance of a shotgun-based microbiome signature using 16S data. Our results demonstrate a reduction in performance when applying the 16S-mapped taxa in the shotgun prediction model, though it retains statistical significance. This suggests that while an exact match between shotgun and 16S data may not yet be feasible, our approach provides a viable method for comparative analysis and validation in the context of CRC-associated microbiome research.

## 1. Introduction

Dysbiosis of the human microbiome plays a critical role in various pathologies and diseases [1,2]. In particular, gut dysbiosis has been linked to colorectal cancer (CRC), which is the world’s third most common cancer and ranks second in mortality [3]. Understanding the microbiome is key to unraveling these widespread diseases and could be a potentially modifiable risk factor. The sequencing of the 16S ribosomal RNA (16S) gene and whole shotgun metagenomic sequencing are the two main current approaches to investigate gut microbiota. 16S may be useful when dealing with a large number of samples, as it offers a balance between cost, speed, and allows abundance estimation of representative bacteria and archaea even with a relatively small number of raw reads [4,5]. However, its taxonomic resolution is often limited to the genus level, though the species-level resolution is improving [6,7]. Furthermore, discordant results may be found when using different primers [6]. Shotgun detects viruses and fungi in addition to prokaryotes, has a higher taxonomic resolution (detects species and, in some cases, even strains of a particular species), and allows for the functional characterization and de novo assembly of new bacterial metagenomes [4]. The downside is its intensive computational demands and the need for substantial sequencing coverage. It also may be less effective when there is a significant presence of host DNA in the sample [6]. 

Due to the great interest aroused by the human microbiome in recent years, a large volume of studies and a large number of data are available to explore host–microbiota associations in health and disease. It is thought that the gut microbiome can play an important role in personalized medicine, for example, in the prediction of some pathologies like CRC [8]. To this end, various machine learning techniques like Random Forest, Logistic Regression (including Lasso), Support Vector Machines, and Artificial Neural Networks have been used to develop prediction models from 16S and/or shotgun taxonomic abundance data [9,10,11]. A primary aim of these studies is to find a “microbial signature” closely associated with the study’s outcome that has high prediction accuracy. The preferred abundance data type in most studies is 16S sequencing, although shotgun use is increasing [8]. A key factor influencing this trend is cost, since despite a decrease in prices, shotgun sequencing is still more expensive than 16S. Currently, only a limited number of studies employ both sequencing technologies. This presents a significant challenge: determining how prediction models and microbial signatures developed using one technology can be adapted for data obtained from the other, given the differing taxonomic resolutions and potential amplification biases, particularly of 16S sequencing. The integration of data from both technologies could leverage the extensive research conducted over the years. In principle, using shotgun data and a 16S model to perform predictions seems more straightforward. For example, in a genus-level 16S model, species-level data from shotgun sequencing can be aggregated by genus before being presented to the model. The reverse case (the input of 16S data into a shotgun-based model) is more challenging. This integration is particularly compelling in clinical settings and routine practices where 16S sequencing remains a more economical option. However, clear criteria for how to incorporate lower-resolution 16S data into a higher-resolution shotgun model are not well established, posing a challenge for effective data integration in these contexts.

We have two objectives in this study. First, we aim to develop an effective one-to-one mapping from shotgun to 16S sequencing data. This mapping is intended to extend the applicability of our previously established Lasso prediction model by Obón-Santacana et al. to also accept 16S data [11]. The model discriminated between CRC patients and healthy controls and was trained from a meta-analysis of eight different published shotgun datasets, with study, age, sex, and Body Mass Index (BMI) as covariates. A robust microbial signature of 32 bacterial species, some of them well established by other studies (e.g., *Parvimonas micra*, *Bacteroides fragilis*), was identified. Our second objective is to evaluate the model’s performance with 16S data. Given the inherently lower resolution of 16S sequencing and the adaptation of a model to a data type for which it was not originally designed, we anticipate a reduction in performance. However, this experiment will provide valuable insights into how much of the model’s predictive power can be retained post-mapping. 

## 2. Results

### 2.1. Description of the Shotgun and 16S Matrices

A validation set of 156 samples (51 controls, 54 high-risk colonic lesions/adenomas, and 51 CRC) was used to estimate the Obón-Santacana et al. model performance. These samples were sequenced again with 16S (see Section 4, Section 4.1 and Section 4.2 for more details). Once we obtained the 16S count matrix, it was subjected to the same pre-processing scheme as the original shotgun data. After filtering rare taxa, the shotgun count matrix retained 469 of the 4027 original taxa, while 16S retained 212 out of 574. Only 30% of the 16S taxa could be identified by name at the species level, but this percentage increased to 76% at the genus level and to 93% at the family level. As shown in Figure 1, 16S abundance data were significantly less diverse than shotgun’s in both richness (Chao1, Shannon) and evenness (Shannon index). Wilcoxon Rank Sum Test between shotgun and 16S alpha diversities gave *p*-values < 2.2 × 10^−16^ for both indices. Also, differences among the control, high-risk lesions, and CRC sample distributions were not apparent. The 16S abundance matrix was clearly sparser, with each sample having on average 61% zeros (Figure 2). In contrast, shotgun samples only had around 4% zeros or less (Wilcoxon Rank Sum Test *p*-value < 2.2 × 10^−16^). We also observed that the control, high-risk lesions, and cancer groups had a similar distribution of zeros in both 16S and shotgun data. The Kruskal–Wallis test did not detect statistically significant differences in the alpha diversity nor the zeros distributions among groups.

A compositional Principal Components Analysis (PCA) based on central log-ratio transformation (clr-PCA) was used to project the 156 samples in a two-dimensional plot (Figure 3). The proportion of variance explained by the 16S’s first and second principal components (PC) was inferior to the analogous PCs in shotgun. The matching between both PCA projections, after accounting for translation, scaling, and rotation effects was assessed with Procrustes analysis, revealing a strong correlation between PCAs: *r* = 0.79 (*p*-value = 0.001). Neither for the shotgun nor for the 16S dataset was an obvious visual clustering of the control, high-risk lesions, and cancer patients achieved.

### 2.2. Taxonomic and Distance-Based Mapping

To use the shotgun-trained model, it is essential that every shotgun taxon in the microbial signature is exclusively mapped to one 16S taxon. To achieve this, the first step was to contrast the 16S taxonomic tree with shotgun’s (details are explained in Section 4, Section 4.5). As shown in Table 1, all 32 bacteria of the shotgun signature could be matched to at least one 16S taxon at the species, genus, family, or order level. Seven species (~22% of the signature) were perfectly matched, and almost 47% had at least one candidate at the genus level. As expected, a greater number of 16S candidates were found in more distantly related taxonomic ranks.

Once the taxonomic matching was complete, nine shotgun taxa (seven at the species level and two at the genus level) were mapped to one specific 16S taxon. The second step was data-driven and concerned only the remaining taxa. The shotgun and 16S clr-transformed abundance matrices were contrasted, so we selected the “closest” 16S species within the pool of candidates as the one with minimum Euclidean distance to the original shotgun species. In this manner, we obtained a group of 16S taxa that can be considered “equivalent” to the original shotgun signature. This equivalence is presented in Figure 4, along with the distance between the shotgun and 16S equivalent taxa. The species with the overall minimum distance is *Parvimonas micra*, which is also the species of the original signature with the greatest contribution to the prediction. In Figure A1 (Appendix A), we show a heatmap representing the clr-transformed abundance matrices for the shotgun microbial signature and the 16S signature side by side. In comparison with the shotgun, the 16S taxa abundances seem more homogeneous across individuals (see also Figure A2b).

### 2.3. Performance of the Mapping in the Validation and Test Sets

The Lasso model by Obón-Santacana et al. was not able to properly discriminate between controls and high-risk lesions but achieved good performance discriminating between controls and CRC cases. The original Area Under the Receiver Operating Characteristic Curve (AUC) of the Lasso model when using the CRC vs. control validation set (102 shotgun samples) was 0.75 (95% CI: 0.66–0.84) [11]. We contrasted this performance to the model’s AUC when using the 16S data for the same 102 samples, and instead of the original microbial signature, we employed the 16S signature presented in Figure 4. With the 16S data, the AUC dropped to 0.64 (95% CI: 0.54–0.75) for CRC vs. controls. The model’s predictions delivered by the shotgun and the 16S data are compared in Figure 5a. Spearman’s ρ between the two is 0.52. The original density plot of the model’s prediction and the 16S density plot are also shown. Obón-Santacana et al. used a threshold of 0.33 that gave a specificity of 0.96, a sensitivity of 0.41, and a precision of 0.91. In the 16S data, the specificity was 0.98, the sensitivity was 0.24, and the precision was 0.92. Following the original paper, we also checked the model’s ability to discriminate between controls and high-risk lesions. As in the case of the shotgun original signature, the 16S mapped signature was uninformative: AUC = 0.52 (95% CI: 0.41–0.64). 

In the final phase of our study, we evaluated the performance of the 16S signature within the Lasso model using an independent 16S test set, comprising 416 samples. This test set was imbalanced, consisting of 39 CRC cases, 146 high-risk lesions patients, and 231 controls. Epidemiological data of this test, contrasted to those of the validation set, are shown in Table 2. We first projected these samples onto a 16S clr-PCA (as shown in Figure 3a) to verify that they were comparable to the validation samples. Although there was an overlap between the two datasets, as seen in Figure A2 in Appendix A, the test samples were notably displaced along the first PC. We then examined the distribution of covariates—sex, age, and BMI—that were used to adjust the original Lasso model, comparing the test set with the validation set. A significant disparity was observed in sex distribution: 51% of the test set were women, while in the validation set, they were only 36%. This is caused by the unbalanced test and a higher proportion of women in the controls (64% vs. 47%). The median age in the test set was also slightly higher, especially in the controls. Due to the test set having a different covariate distribution, AUC was adjusted by the covariates following Pepe and Cai’s analysis of placement values [12]. A confidence interval (CI) was computed using 2000 bootstrap resamples of the model prediction. Therefore, the AUC we obtained for the 16S test set was 0.61 (95% CI: 0.51–0.71). Receiver Operating Characteristic (ROC) curves for shotgun, 16S validation, and 16S test are shown in Figure 5b. At the threshold of 0.33, we obtained a specificity of 0.97, a sensitivity of 0.10, and a precision of 0.36. We also computed the performance for the controls vs. high-risk lesions for this test set, and again, we obtained an AUC of 0.52 (95% CI: 0.45–0.58).

## 3. Discussion

It is well known that results derived from shotgun and 16S sequencing technologies are not easy to reconcile [13]. Problems like disparate taxonomic resolution, potential biases of the 16S amplification, and differing reference databases may produce very different abundance matrices, PCAs, prediction models, and/or relevant microbial biomarkers. In the present study, we describe an algorithm to map taxa from shotgun to 16S. Furthermore, we show that replacing a shotgun model’s microbial signature with the 16S taxa selected by this mapping decreases the model’s AUC, though the model still performs better than chance alone. In our case, sensitivity and precision were also lower at the original model’s threshold, while specificity was unaffected. To our knowledge, previous cancer-control studies with shotgun and 16S data available trained two separate models and often noted that shotgun had slightly better performance (see [14,15] about virome in CRC and [16] in pancreatic cancer) but did not try to assess the shotgun model accuracy when predicting 16S data. Our mapping approach combined taxonomic and data-driven approaches, selecting the “nearest” 16S taxa to the shotgun microbial signature but always prioritizing biological coherence. Overfitting did not seem to be a major problem since the procedure was agnostic to the outcome. Our approach is also valid for unsupervised analyses, for instance, to project 16S data over a shotgun PCA (see Appendix A, Figure A2), which may be of interest when clear clusters are observed. 

Not all taxa in the microbial signature could be mapped to 16S with the same accuracy. However, we found that most of the species already highlighted in other studies had a good shotgun–16S correspondence. For instance, *Parvimonas micra* and *Bacteroides fragilis* have been consistently associated with CRC in a wide range of studies and cohorts [17]. The former is also the species with greatest importance in Obón-Santacana et al.’s model [11]. In our data, we have found that the profile of both bacteria across the 156 samples is very similar in the shotgun and 16S abundance matrices. Not only are these species identified by name and present in both taxonomic tables, but as we show in Figure 4, they have the absolute minimum distance. Other species with a low shotgun–16S distance and that have been associated with CRC discrimination in the literature were *Bifidobacterium bifidum*, *Faecalibacterium prausnitzii* (the second most important species in the prediction model), and *Dialister invisus*. On the other hand, *Sutterella wadsworthensis* (ranked third in the model, though scarcely found in the CRC literature) presents an abundance profile in 16S that is very different from that of shotgun, especially regarding the cancer samples (see Figure A1).

The mapping algorithm we propose also has its own set of drawbacks and challenges. Firstly, shotgun and 16S should be pre-processed in a similar way to make them comparable at the taxonomic and abundance matrix levels. This is not easy since shotgun and 16S data have different particularities. For instance, a legitimate question is whether it is appropriate to impose the shotgun filtering criteria (as we did) when 16S is clearly sparser and less diverse. Also, not only does shotgun have greater resolution, but taxonomies are also vastly different due to the different reference databases of shotgun and 16S and the frequent update of the microbial phylogenies. We tried to alleviate these issues using a mixed taxonomic/data-driven nature, but that requires a fraction of microbiome samples sequenced with 16S and shotgun, which is not the most common scenario. Also, the best metric to map shotgun to 16S data should be decided by the researcher and may change depending on the dataset or the problem at hand. We opted for the Euclidean distance because it is easy to compute and interpret, but the mapping was computed taxon by taxon and does not consider potential interactions between the bacteria. Increasing the number of sequenced 16S regions might have improved the taxa resolution. Another limitation in our study that probably reduced the mapping efficiency was that the shotgun database used (UHGG v1.0) was outdated. This was related to our interest in validating a predictive model that had been developed with that version. Finally, although we successfully mapped the shotgun’s signature to 16S, the decrease in the model’s AUC was considerable (0.75 to 0.64 in the validation set and 0.61 in the test set). A possible explanation is that we were restricted to only 156 patients that had both shotgun and 16S data: with a larger paired sample, the quality of the shotgun to 16S mapping may increase. Also, a larger sample of patients would allow for retraining the model’s coefficients and potentially improve the results. 

In summary, finding 16S taxa that are “equivalent” to shotgun taxa is possible but still challenging, and several preconditions should be met. Recent strategies like Greengenes2 [13] are promising, as the use of the same reference database for both kinds of data allows a more unified result from the bioinformatics step. In the other cases, our contribution may be useful to place shotgun-generated and 16S-generated data, prediction models, and microbial signatures in a common ground. 

## 4. Materials and Methods

### 4.1. Study Population and Design

The research cohort (COLSCREEN study) was recruited among individuals that participated from 2016 to 2020 in the ongoing population-based CRC screening program overseen by the Catalan Institute of Oncology in L’Hospitalet del Llobregat, Barcelona, Spain [11]. This program invites men and women between the ages of 50 and 69 to partake in the immunochemical fecal occult blood test (FIT). In the event of a positive FIT result (≥20 μg Hb/g feces), it is recommended that the participants undergo colonoscopy. Participants of the COLSCREEN study (N = 997) were invited to participate after receiving a positive FIT result. Since CRC diagnosis is rare in screening, this cohort includes patients diagnosed clinically from the CRC Functional Unit (N = 45). Furthermore, a subset of participants with a negative FIT is also included (N = 140). All of them underwent a colonoscopy, and participants were categorized based on the risk-stratification proposal by Castells et al. [18] after a careful review of the colonoscopy and histopathology reports.

A subset of the patients consisting of 156 individuals selected from the COLSCREEN study (51 controls with normal colon mucosa, 54 with high-risk precancerous lesions, and 51 with CRC) were previously used to validate a predictive model for CRC proposed by Obón-Santacana et al. [11]. The model was trained with meta-analysis data from eight different published shotgun datasets, with study, age, sex, and BMI as covariates. For testing the model in an independent set, the remaining patients available in the COLOSCREEN study with CRC (N = 39), with high-risk lesions (N = 165), and controls (N = 231) were used (total N = 416).

All participants who agreed to take part in the COLSCREEN study provided written informed consent, donated a fecal and blood sample at recruitment (samples obtained before colonoscopy), and answered an epidemiological questionnaire. Clinically diagnosed CRC patients usually underwent a colonoscopy before recruitment, and the fecal samples were obtained prior surgery. In the present study, we excluded those participants that reported having used antibiotics or probiotics one month before sampling. The ethics committee of the Bellvitge University Hospital, L’Hospitalet del Llobregat, Barcelona, Spain, approved the protocol of the study (PR084/16), and all procedures were performed in accordance with relevant guidelines and regulations.

### 4.2. DNA Extraction, Sequencing, and Bioinformatics Analysis

Though the stool samples used for shotgun and 16S sequencing were identical, the DNA extractions were performed separately for each method. The shotgun sequencing process has been detailed previously [11]. In summary, fecal DNA was extracted using the NucleoSpin Soil Kit (Macherey-Nagel, Duren, Germany) following the manufacturer’s protocol. Sequencing libraries were prepared with 2 µg of total DNA using the Nextera XT DNA Sample Prep Kit (Illumina, San Diego, CA, USA). The sequencing was performed with 150 nucleotides, paired-end, using an Illumina HiSeq 4000 platform. Human reads were removed from the metagenome samples by aligning the reads to the human genome (GRCh38) with Bowtie2. Reads were deduplicated and trimmed to remove sequencing adapters and low-quality ends. Clean sequencing reads were classified using Kraken2 (v.2.1.0) with a filtering threshold of 0.1, followed by Bayesian reassignment at the species level using Bracken2. The UHGG database v.1.0 [19] was used for this classification.

The 16S rRNA sequencing was performed on all 997 COLSCREEN samples following a standard protocol for stool. DNA extraction was performed using the DNeasy PowerLyzer PowerSoil Kit (Qiagen, Venlo, The Netherlands, ref. QIA12855), including negative controls of extraction. The extracted DNA was used to prepare 16S rRNA libraries, targeting the V3-V4 region of the bacterial 16S ribosomal RNA gene, using the following universal primers in a limited-cycle PCR: V3-V4-Forward (5′-TCGTCGGCAGCGTCAGATGTGTATAAGAGACAGCCTACGGGNGGCWGCAG-3) and V3-V4-Reverse (5′-GTCTCGTGGGCTCGGAGATGTGTATAAGAGACAGGACTACHVGGGTATCTAATCC-3′). Then, full-length Nextera adapters with barcodes for multiplex sequencing were added in a second PCR step, resulting in sequencing-ready libraries. Sequencing was performed in the Illumina MiSeq with 2 × 300 bp reads using v3 chemistry. Two bacterial mock communities from the BEI Resources of the Human Microbiome Project (HM-276D and HM-277D) were amplified and sequenced in the same manner as all other samples. Negative controls of PCR amplification were also included in parallel, using the same conditions and reagents.

Raw data were processed using the Dada2 pipeline (v. 1.12.1) [20]. We filtered and trimmed out low-quality reads according to the observed quality profiles. The value for maximum expected error was 2. Also, 10 reads from the start of each read were removed. Identical sequencing reads were combined into unique sequences, and then we made a sample inference from a matrix of estimated learning errors and merged paired reads. After the removal of chimeric sequences, taxonomy was assigned utilizing the SILVA 16S rRNA database (v.132) [21]. The output of the process consisted of a count matrix (sample by microbial taxa) and a taxonomic table (lineage of each microbial taxa).

As the reference databases are different in shotgun and in 16S data (UHGG v.1.0 for the former and SILVA v.132 for the latter), in some cases, there are incongruences in the taxonomic assignment. The same microorganism may appear under a different name in shotgun and 16S; even the full lineage may be affected in some cases. To ensure the comparability of both taxonomy tables, we standardized them to follow the NCBI taxonomic nomenclature (date: 7 March 2023) using the *taxonomy* function from the R package myTAI (v-0.9.3) [22]. Each unique sequence name for 16S taxa obtained from SILVA database was mapped to the NCBI, with a success rate of 864/948 (91%). The 84 taxa not found by the *taxonomy* search were manually curated.

### 4.3. Pre-Processing of the Abundance Matrices

Shotgun count matrix was normalized by genome length. From this point, both count matrices were subject to the same pre-processing scheme presented in the original paper. First, we dropped all the species that were not present in (at least) 5% of the samples with 0.1% abundance or higher. We computed the percentage of zeros for each sample for further comparison between both tables. Then, we performed a replacement of zero values (using the square root Bayesian-Multiplicative method) followed by clr transformation with the *zCompositions* (v1.4.0) R package [23].

### 4.4. Description of the Abundance Matrices

The pre-processed shotgun abundance matrix was the same as the one used for validating the Lasso model. We performed several descriptive analyses for the shotgun and 16S matrices. Shannon and Chao1 alpha-diversity indices were computed from the filtered data prior to the zero-substitution step. In addition, to compute the Shannon index, we rarified the filtered data to the minimum depth. All alpha-diversity analyses were performed using the *vegan* (v2.6) R package [24]. A clr-PCA was used to project graphically the Shogun abundance data, on the one hand, and the 16S on the other. Then, we used the *vegan* (v2.6) Procrustes analysis to search for the rotation of maximum agreement between the PCAs and computed the sum-of-squared errors and Procrustes correlation between the same samples in both projections.

### 4.5. Mapping Shotgun to 16S Abundance Data

Mapping data from the two sequencing technologies requires establishing a correspondence between the taxa identified using 16S and those identified using shotgun sequencing. Moreover, to use a shotgun-based model, we need every shotgun taxon to be mapped to a single 16S taxon. To achieve this one-to-one mapping from shotgun to 16S data, we used a two-step approach: taxonomic mapping and data-driven mapping.

Taxonomic: Here, we compared the taxonomic trees of shotgun and 16S and searched the latter for the species of interest. In our case, the first step involved checking whether the 32 species in the Lasso model’s signature were among the 16S taxa. If a direct match was not found, the algorithm was extended to higher taxonomic ranks (such as family or order) until a match was identified. This taxonomic strategy is “universal” in that it only requires the availability of taxonomic tables for both 16S and shotgun sequencing, which is generally the case. Notably, the datasets for shotgun and 16S sequencing do not need to be paired; they can originate from different individuals. However, this approach faces several challenges and limitations. Firstly, it requires that taxa lineages be identical in both phylogenetic trees, meaning that the same microorganism should be classified with the same species name and lineage in both datasets. This is often not the case due to rapid updates in microorganism phylogeny and variations in reference databases. As a result, taxa classification must be standardized to the same nomenclature in both shotgun and 16S datasets. Secondly, the lower resolution of 16S sequencing means that many taxa identified at the species level in the shotgun may not be present in the 16S dataset, or only identifiable at the genus level or higher. Consequently, a single taxon identified in shotgun sequencing could correspond to multiple candidate taxa in the 16S dataset. In those cases, we proceeded with the second step.Data-driven: For this second step, it is essential to have paired samples, i.e., samples that are sequenced using both shotgun and 16S techniques. Then, some metrics may be devised to select the “closest” 16S taxon to a particular shotgun taxon in a data-driven way. We propose computing the Euclidean distance between relevant shotgun taxa and all 16S taxa using transposed abundance matrices where samples are treated as variables. The chosen 16S taxon is the one with the “closest” clr-transformed abundance profile across all samples to the target shotgun species. The main advantage of this approach is that no information about bacterial phylogeny is needed. The disparate taxonomic resolutions of the shotgun and 16S sequencing techniques are by-passed; in fact, knowing the species or genus name (or even their lineage) is not mandatory. However, if the sample size is limited, this method may face difficulties in obtaining significant separation of 16S taxa and result in wrong mappings due to perceiving noise in the abundance data as meaningful variation. The shotgun model’s performance using these 16S biomarkers may be misleadingly optimistic; thus, using an independent test set (additional samples sequenced with 16S) is advisable to estimate the true performance.

### 4.6. Performance Evaluation

We evaluated the Lasso predictive model using the mapped 16S taxa (the “closest” taxa to the shotgun original microbial signature) as features. To do so, we assessed the performance of the original validation set (N = 156) when using 16S data, as well as the correlation between the original shotgun prediction and the current prediction. For the present study, we exclusively computed the AUC for two different comparisons: (a) the 51 control (defined as normal/no-lesions) vs. 51 COLSCREEN CRC cancer samples and (b) the 51 controls vs. 54 high-risk colonic precancerous lesions. This approach aligns with the original Lasso model, which was designed for binary prediction. Finally, we evaluated the model performance when using a 16S independent test set (N = 416) that consisted of 231 controls, 39 CRC patients, and 146 high-risk lesions. For this test set, we had 16S abundance data, sex, age, and BMI information—the same covariates used in adjusting the Lasso model.

## Figures and Tables

**Figure 1 ijms-25-01181-f001:**
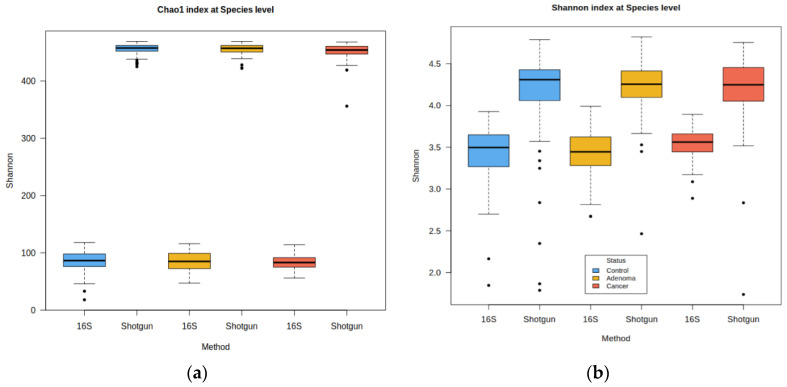
Shotgun vs. 16S alpha diversity. Controls are in blue, high-risk samples in yellow, and colorectal (CRC) samples in red. (**a**) Chao1 index: Kruskal–Wallis test among the three diagnostic groups: shotgun *p*-value = 0.991, 16S *p*-value = 0.152; (**b**) Shannon index: Kruskal–Wallis: shotgun *p*-value = 0.152, 16S: *p*-value = 0.732.

**Figure 2 ijms-25-01181-f002:**
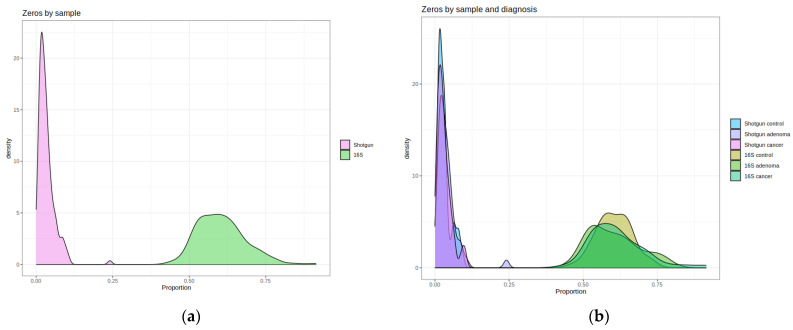
Shotgun vs. 16S sparsity. As the number of taxa differs between the two matrices, the proportion of zeros was computed for each sample. (**a**) The proportion of zeros in shotgun (purple) and 16S (green): Wilcoxon Rank Sum Test *p*-value < 2.2 × 10^−16^; (**b**) the proportion of zeros in control, high-risk lesions, and cancer for shotgun (blue-purple) and 16S (brown-green): Kruskal–Wallis among the three diagnostic groups: shotgun *p*-value = 0.152, 16S *p*-value = 0.154.

**Figure 3 ijms-25-01181-f003:**
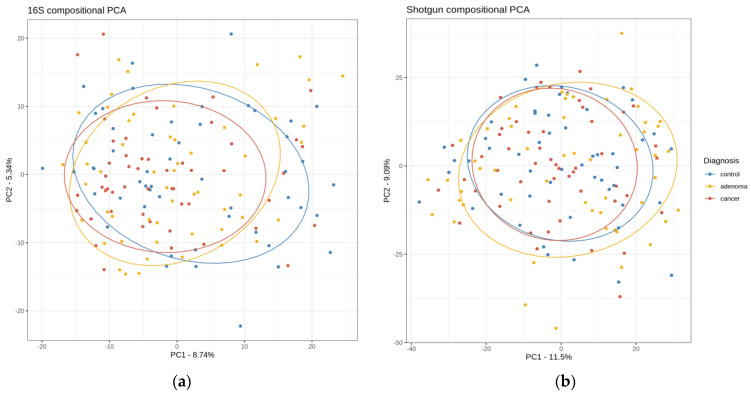
Shotgun and 16S clr-PCA of the 156 validation samples. Controls are in blue, high-risk samples are in yellow, and CRC samples are in red. (**a**) 16S data; (**b**) Shotgun data. Procrustes *r* between both PCAs was 0.79 (*p*-value = 0.001). Axes show the percentage of variance explained.

**Figure 4 ijms-25-01181-f004:**
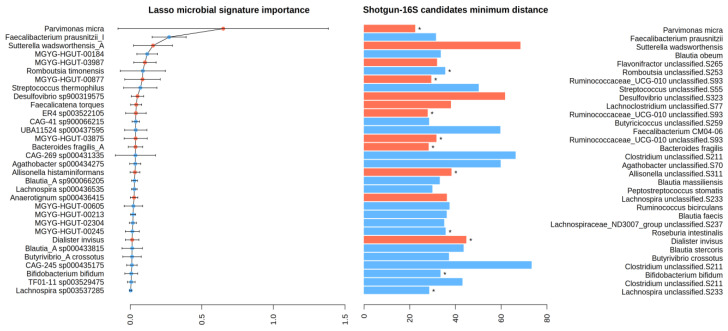
Distance-based matching between shotgun and 16S candidates. On the left, we show the 32 shotgun taxa that constitute the original bacterial signature, while on the right, we present their 16S counterparts, which were chosen after the taxonomic and distance-based matching. To assess the impact of each shotgun species in the Lasso model, they are sorted in descending order by their importance, i.e., their coefficient in the Lasso prediction model multiplied by their absolute average abundance (error bars stand for the abundance standard deviation). Blue and red mean that a species is either control- or CRC-enriched, respectively (see also Figure 3 in [11]). Bars correspond to the Euclidean distance between the shotgun species and their mapped species in the 16S dataset. * marks eleven species whose distance is also the absolute minimum when all 16S taxa are considered; i.e., they are the closest species even if the previous step (the taxonomic matching) is omitted.

**Figure 5 ijms-25-01181-f005:**
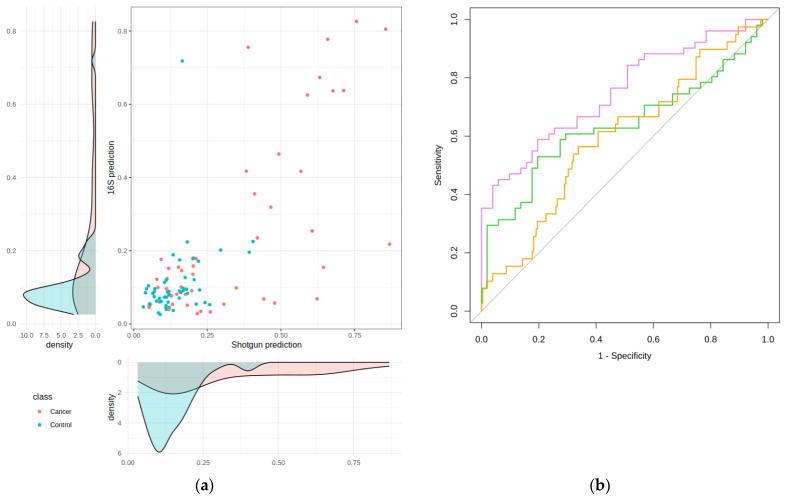
(**a**) Shotgun vs. 16S predictions (validation set); Spearman’s *ρ* = 0.52. Samples below the threshold value of 0.33 are assigned to the control group (blue), while those above this value are predicted to belong to patients with CRC (red). The density plot of shotgun prediction and the 16S density plot are shown as marginals; (**b**) ROC curves for the original shotgun validation data (purple), 16S validation data (green), and 16S test data (orange).

**Table 1 ijms-25-01181-t001:** Taxonomic matching between shotgun and 16S. We show which taxa of the shotgun bacterial signature could be matched to a species, genus, family, or order present in the 16S taxa, the frequency of assignments to each taxonomic rank (in absolute and relative numbers), and the median and range number of candidate taxa by rank.

	Species	Genus	Family	Order
Bacterial signature (original Lasso model [11])	*Bacteroides fragilis A*	*Agathobacter sp000434275*	*Anaerotignum sp000436415*	*Lachnospira sp000436535*
	*Bifidobacterium bifidum*	*Allisonella histaminiformans*	*CAG-41 sp900066215*	*MGYG-HGUT-03875*
	*Butyrivibrio A crossotus*	*Blautia A sp000433815*	*ER4 sp003522105*	
	*Dialister invisus*	*Blautia A sp900066205*	*Faecalicatena torques*	
	*Faecalibacterium prausnitzii I*	*CAG-245 sp000435175*	*MGYG-HGUT-00877*	
	*Parvimonas micra*	*CAG-269 sp000431335*	*MGYG-HGUT-02304*	
	*Sutterella wadsworthensis A*	*Desulfovibrio sp900319575*	*MGYG-HGUT-03987*	
		*Lachnospira sp003537285*	*TF01-11 sp003529475*	
		*MGYG-HGUT-00184*		
		*MGYG-HGUT-00213*		
		*MGYG-HGUT-00245*		
		*MGYG-HGUT-00605*		
		*Romboutsia timonensis*		
		*Streptococcus thermophilus*		
N (% of total)	7 (21.9%)	15 (46.9%)	8 (25%)	2 (6.2%)
Median (Range) number of candidates	1 (0)	6 (1–68)	47 (14–68)	87.5 (2–173)

**Table 2 ijms-25-01181-t002:** Summary of sample sizes and epidemiological data of validation and test sets, stratified by diagnosis.

	Diagnosis	N	Woman (%)	Age Median (IQR)	BMI Median (IQR)
Validation	Controls	51	47.1%	57 (7)	26.2 (4.4)
	High-risk lesions	54	33.3%	60 (9.5)	28.1 (5.6)
	CRC cases	51	27.5%	65 (13.5)	26.9 (4.3)
	**Total**	**156**	**35.9%**	**60 (7.9)**	**27.1 (4.2)**
Test	Controls	231	64.1%	60 (10)	27.1 (6.7)
	High-risk lesions	146	35.6%	63 (7)	27.7 (4.9)
	CRC cases	39	25.6%	66 (6.5)	27.5 (5.8)
	**Total**	**416**	**50.5%**	**62 (6.0)**	**27.4 (5.1)**

## Data Availability

The taxonomic and abundance data of the 997 fecal samples are available at the Zenodo repository, https://zenodo.org/records/10376600. The previously published 156 Shotgun samples can be found at https://zenodo.org/records/6671562 (both accessed on 14 December 2023). Raw data can be found at the European Nucleotide Archive (ENA) under project PRJEB71787.

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
