# Peer review of "Performance of a Shotgun Prediction Model for Colorectal Cancer When Using 16S rRNA Sequencing Data"

_ijms, 2024, doi:10.3390/ijms25021181_

Round 1

Reviewer 1 Report

Comments and Suggestions for Authors

Manuscript by Ramon et al., describes the extension of a ML approach to discriminate among 3 classes based on Shotgun data to DNA-metabarcoding data. It is well written and the scope of the research is well described in the introduction.

Nonetheless, there are some concerns the authors should address. Principally, the taxa mapping from 16S data to shotgun is not completely clear: 

·       While in the manuscript is described how 16S data were analysed, a description about the shotgun data analysis is missing. Even if it should be described in Obo-Santacan et al., a short note needs to be added also here. 

·       Regarding the classification of DNA-metabarcoding data authors have used an outdated database, and this choice might negatively influence the taxon mapping among the two approaches. Moreover, it is not clear how the taxonomic classification was performed. For instance, have the authors used a Bayesian classifier or a classifier relying on sequence similarity? Another issue the author should consider, is related to the fact that classification at species level by using a short sequence are not reliable.

·       In lines 351-353 authors stated the standardized both the shotgun and 16S data to the NCBI taxonomy. Actually a through description of this step is missing.

·       L 373-413: In my opinion mapping on the taxonomy is influenced by technical limitation of both the applied techniques. Moreover, considering the intra-genomic variability of 16S sequences, taxonomic classifications at different taxonomic levels may be obtained from ASVs belonging to the same genome. This may affect the euclidean based taxon mapping. Indeed, the distances observed in figure 4 seems to be too large to effectively asses the mapping worked correctly. Probably different approaches should be considered, such as phylogenetic placing.

I cannot find any release of neither raw sequencing data nor metadata. 

Author Response

Manuscript by Ramon et al., describes the extension of a ML approach to discriminate among 3 classes based on Shotgun data to DNA-metabarcoding data. It is well written, and the scope of the research is well described in the introduction. Nonetheless, there are some concerns the authors should address. Principally, the taxa mapping from 16S data to shotgun is not completely clear:

  • While in the manuscript is described how 16S data were analysed, a description about the shotgun data analysis is missing. Even if it should be described in Obón-Santacana et al., a short note needs to be added also here.

Thank you for rising this point. We have revised the methods section to include a summary of the shotgun analytical pipeline.

  • Regarding the classification of DNA metabarcoding data authors have used an outdated database, and this choice might negatively influence the taxon mapping among the two approaches. Moreover, it is not clear how the taxonomic classification was performed. For instance, have the authors used a Bayesian classifier or a classifier relying on sequence similarity? Another issue the author should consider, is related to the fact that classification at species level by using a short sequence are not reliable.

We are aware that the database used for the shotgun sequencing is now outdated, since UHGG v.2.02 is available at the EBI website. Since the aim of the study was to validate a model that had been trained with version 1.0 of the database, we intentionally did not change the shotgun model. This is a limitation that we now have acknowledged in the discussion. Probably updating the reference database might improve the mapping of shotgun to 16S. However, this would have required retraining the original model and probably a different one would result. The more recent publication of Greengenes2 that provides a 16S database built from whole genomes should also improve the mapping. However, since most of the 16S studies recently published have used the DADA2 pipeline with the SILVA reference and show the difficulties of translating shotgun findings to 16S.

  • In lines 351-353 authors stated the standardized both the shotgun and 16S data to the NCBI taxonomy. Actually a thorough description of this step is missing.

We used R package myTAI to retrieve NCBI terms. We have included this in the methods section to describe the procedure used to standardize to the NCBI taxonomy.

  • L 373-413: In my opinion mapping on the taxonomy is influenced by technical limitation of both the applied techniques. Moreover, considering the intra-genomic variability of 16S sequences, taxonomic classifications at different taxonomic levels may be obtained from ASVs belonging to the same genome. This may affect the Euclidean based taxon mapping. Indeed, the distances observed in figure 4 seems to be too large to effectively assess the mapping worked correctly. Probably different approaches should be considered, such as phylogenetic placing.

We agree that the mapping of the taxonomy is the most challenging task, mostly because of the low resolution of 16S and the different reference databases between shotgun and 16S data. After facing the fact that only 7 out of 32 taxa could be directly mapped, we moved forward to a data-dependent approach based on distances. Euclidean distances are difficult to interpret in absolute value because depend on the sample size. Figure 4 shows that most taxa were at distances between 20-40, but 7 were around 60. The fact is that taxa Sutterella wadsworthensis, which was mapped by taxonomic placement, showed a distance in the higher end, which we interpret as evidence that the data-driven mapping overall was reasonable. We acknowledge that phylogenetic placing could be an alternative approach that we have not tried and might improve the mapping.

I cannot find any release of neither raw sequencing data nor metadata.

Raw sequencing data will be deposited in ENA repository under project PRJEB71787. We have registered the project and are in the process of uploading the fastq files, which will be made available upon acceptance of the manuscript. The counts and metadata was already deposited in zenodo.

Reviewer 2 Report

Comments and Suggestions for Authors

Dear Authors,

thank you for preparing this interesting manuscript. It is well written and good to follow. Please find my comments below:

Line 206: Please set a “.” after 231 controls

Line 349: Did you try combining two gene banks (for example Greengene and Silva) to get more taxa for 16S?

Appendix A: AL4084 and AL4085 are in the same line. Is this correct? As this should be one patient per line in my understanding something might be wrong, maybe the resolution. Please check.

Do you think that including more regions for 16S could improve the results and so your model (V2, V5,V6 for example)?

A happy new year to all of you!

kind regards

Author Response

thank you for preparing this interesting manuscript. It is well written and good to follow. Please find my comments below:

  • Line 206: Please set a “.” after 231 controls

Thank you, we have fixed this typo and revised the whole document.

  • Line 349: Did you try combining two gene banks (for example Greengenes and Silva) to get more taxa for 16S?

We only used Silva database, which is the most frequently used for 16S when the DADA2 pipeline. We agree that the recent release of Greengenes 2.0 may prove a better approach for future studies that aim to combine 16S and shotgun data, that was mentioned in the discussion. The combination of diverse references is challenging, since from the same raw data, slightly different results are obtained and merging the results would be very difficult.

  • Appendix A: AL4084 and AL4085 are in the same line. Is this correct? As this should be one patient per line in my understanding something might be wrong, maybe the resolution. Please check.

Thanks, the label is correct. The fact is that the figure was produced with R and when many rows are shown in the heatmap, not all the labels are printed. There are two samples that were sequenced twice as part of the quality control and the reads merged. We have maintained the original lab sample IDs for these labels. Only one of these labels appear in the figure, AL4084AL4085. The sample file uploaded to Zenodo also includes sample AL4026AL4052.

  • Do you think that including more regions for 16S could improve the results and so your model (V2, V5,V6 for example)?

It is expected that more 16S regions improves the resolution, but common practice is to use V3 and V4 only. We have included this in the discussion.

Round 2

Reviewer 1 Report

Comments and Suggestions for Authors

authors have addressed all my concerns.